# Allergy to Peanuts imPacting Emotions And Life (APPEAL): The impact of peanut allergy on children, teenagers, adults and caregivers in the UK and Ireland

Marina Tsoumani[1], Lynne Regent[2], Amena Warner[3], Katy Gallop[4], Ram Patel[5], Robert Ryan[6], Andrea Vereda[6], Sarah Acaster[4], Audrey DunnGalvin[7]*, Aideen Byrne[8]

1 Division of Infection, Immunity and Respiratory Medicine, School of Biological Sciences, NIHR Manchester Biomedical Research Centre, Manchester University Hospitals NHS Foundation Trust, Manchester Academic Health Science Centre, University of Manchester, Manchester, United Kingdom, 2 Anaphylaxis Campaign, Farnborough, United Kingdom, 3 Allergy UK, Kent, United Kingdom, 4 Acaster Lloyd Consulting, London, United Kingdom, 5 Brainsell Ltd., London, United Kingdom, 6 Aimmune Therapeutics, A Nestlé Health Science Company, London, United Kingdom, 7 School of Applied Psychology and Department of Paediatrics and Child Health, University College Cork, Cork, Ireland, 8 Allergy Dept, Children's Health Ireland and Paediatrics and Child Health Trinity College Dublin, Dublin, Ireland

* A.DunnGalvin@ucc.ie

**Data Availability Statement:** Data relevant to this study are reported in the article. The APPEAL study

## Abstract

The **A**llergy to **P**eanuts im**P**acting **E**motions **A**nd **L**ife study (APPEAL) explored the psychosocial burden of living with self-reported peanut allergy experienced by children, teenagers, adults and caregivers in the UK and Ireland. A two-stage (quantitative survey and qualitative interview [APPEAL-1]), cross-sectional study of the psychosocial burden of peanut allergy (APPEAL-2) was conducted. Quantitative data were evaluated using descriptive statistics and qualitative data were analysed using MAXQDA software. A conceptual model specific to UK and Ireland was developed using the concepts identified during the analysis. A total of 284 adults in the UK and Ireland completed the APPEAL-1 survey and 42 individuals participated in APPEAL-2. Respondents reported that peanut allergy restricts their choices in various situations, especially with regard to choosing food when eating out (87% moderately or severely restricted), choosing where to eat (82%), special occasions (76%) and when buying food from a shop (71%). Fifty-two percent of survey participants and 40% of interview participants reported being bullied because of PA. Psychological impact of peanut allergy included feeling at least moderate levels of frustration (70%), uncertainty (79%), and stress (71%). The qualitative analysis identified three different types of coping strategies (daily monitoring or vigilance, communication and planning) and four main areas of individuals' lives that are impacted by peanut allergy (social activities, relationships, emotions and work [adults and caregivers only]). The extent of the impact reported varied substantially between participants, with some reporting many negative consequences of living with peanut allergy and others feeling it has minimal impact on their health-related quality of life. This large survey and interview study highlight the psychosocial burden of peanut allergy for adults, teenagers, children and caregivers in the UK and Ireland. The analysis demonstrates the wide

data used in this study can be found at the provided references, 19–22.

**Funding:** This work was supported by Aimmune Therapeutics. Authors Robert Ryan and Andrea Vereda are employed by Aimmune Therapeutics. Brainsell Ltd received funding from Aimmune Therapaeutics for conducting the APPEAL-1 survey and for analytical support. Author Ram Patel is employed by Brainsell. Acaster Lloyd Consulting received funding from Aimmune Therapeutics for conducting the study. Authors Katy Gallop and Sarah Acaster are employed by Acaster Lloyd Consulting. The specific roles of these authors are articulated in the 'author contributions' section. The funders provided support in the form of salaries to these authors or consultant fees to the employers of these authors and were involved in the study design, data collection and analysis, the decision to publish, and preparation of the manuscript.

**Competing interests:** The authors have read the journal's policy and have the following competing interests: RR and AV are employed by Aimmune Therapeutics. SA, KG, and RP report consulting for Aimmune Therapeutics. MT, LR, AW, AD, and AB held consultancy and received sponsorship to attend and speak at educational events from pharmaceutical companies. This does not alter our adherence to PLOS ONE policies on sharing data and materials.

variation in level of impact of peanut allergy and the unmet need for those individuals who experience a substantial burden from living with peanut allergy.

## Introduction

In the UK and Ireland peanut allergy (PA) is the third most common food allergy, affecting between 0.5–2.5% of children [1–5] and up to 0.5% of adults [6]. Egg and milk allergy resolve for the majority after early childhood, leaving peanut as the most common persistent food allergy. Individuals with PA have to strictly avoid peanuts or food containing peanuts and carry emergency medication for use if accidental exposure occurs, and there is not yet an approved treatment for PA in the UK or Ireland.

There is a growing body of evidence of the negative impact of food allergy on the health-related quality of life (HRQL) of individuals with food allergy and their caregivers [7–10]; however, the evidence relating specifically to PA is more limited. Research explicitly on PA is valuable to assess whether there is a similar impact to other food allergies, particularly because for the majority of individuals, PA is lifelong [11–14].

Recent research using validated instruments (Food Allergy Quality of Life Questionnaire; Food Allergy Independent Measure; EQ-5D) to assess the impact of PA on the HRQL of children and teenagers in the UK found that caregivers report high levels of psychosocial burden for their child, particularly those with more severe PA and a recent history of reactions. Male and female children were equally impacted [15]. The same authors [16] also explored the impact of caring for a child with PA on caregivers, using validated instruments such as the EQ-5D and Hospital Anxiety and Depression Scale [17]. The study found that male and female caregivers reported significantly higher anxiety than UK population norms, with 26% of male and 35% of female caregivers reporting probable clinical anxiety, compared to UK norms (12.5% of males; 19% of females) [18]. Caregivers of children with severe PA reported significant impact on their employment and on HRQL. Caregiver gender did not impact the level of burden experienced, therefore indicating that male caregivers are equally as anxious and experience the same level of HRQL, career and productivity impact as female caregivers [16].

The APPEAL (Allergy to Peanuts imPacting Emotions And Life) study was conducted to explore the psychosocial burden of living with PA in Europe. The APPEAL study design consisted of two phases (a quantitative survey and qualitative interviews) conducted across eight European countries (Germany, UK, France, Spain, Denmark, Ireland, Italy and the Netherlands). The results of the study overall have been published [19–21]; however, this article focuses on the results collected from participants in the UK and Ireland only because they have similar cultures, food habits, and healthcare systems. Given the lack of UK and Ireland evidence relating to PA rather than food allergies in general, this article highlights the burden experienced by children, teenagers, adults and caregivers in the UK and Ireland to this most common of persistent food allergies.

## Methods

### Study design

The APPEAL study was a two-stage, cross-sectional study of the psychosocial burden of PA conducted in eight European countries. Details of the full APPEAL study have been previously reported [19–21]. This article focuses on the results from participants in the UK and Ireland.

## Procedures

The first stage of the study (APPEAL-1) consisted of a bespoke online survey designed to assess the burden and impact of PA on individuals with PA and their caregivers. The survey can be found as a supplement to an earlier APPEAL publication [22]. The APPEAL survey was developed by the APPEAL advisory board, which included representatives of eight patient advocacy groups (one from each of the eight countries in which the study was conducted) and a specialist panel of five healthcare professionals and research specialists. For most survey questions, a five-point response scale was used (in general, 1 indicated the lowest impact/better HRQL and 5 highest impact/worse HRQL). The questionnaire included demographics and clinical characteristics, practical issues of PA management and psychosocial impacts.

The second stage (APPEAL-2) consisted of in-depth telephone or in-person interviews with children (aged 8–12 years), teenagers (aged 13–17 years) and adults (aged 18–30 years) with PA, and adult caregivers to children (aged 4–17 years) with PA. The interviews were conducted by researchers trained to conduct qualitative interviews.

All interviews with children were conducted in person; parents/caregivers were not present during child interviews and vice versa. The interview guide allowed participants to spontaneously describe how PA affects them in addition to using pre-specified probes if concepts were not raised. Caregivers of children and teenagers (aged 4–17 years) with a diagnosis of PA were asked about the impact of PA on their child with PA, as well as on their own life. The guide is available as a supplement to the original APPEAL-2 publication [21]. All interviews were recorded and transcribed verbatim, with any identifying data (eg, reference to a name) removed.

## Participants

Two independent samples of participants were recruited for APPEAL-1 and APPEAL-2. APPEAL-1 participants were recruited through patient advocacy groups and specialist patient recruitment panels; APPEAL-2 participants were all recruited through specialist patient recruitment panels that engaged participants from databases of individuals willing to participate in research studies.

Participants were eligible for APPEAL-1 if they were adults with a self-reported diagnosis of PA or a caregiver to an individual of any age with a diagnosis of PA, were a resident of the UK or Ireland, and had not taken part in a market research study on PA in the previous two months.

Participants were eligible for APPEAL-2 if they were a child (aged 8–12 years), teenager (aged 13–17 years) or adult (aged 18–30 years) with self-reported moderate or severe PA (PA diagnosed by a medical doctor) or a caregiver of a child aged 4–17 years with moderate or severe PA, and a resident of the UK or Ireland. Participants were excluded from APPEAL-2 if they had never experienced a reaction to peanut in their day-to-day life (eg, only as a result of a provocation test using food challenge); recruitment aimed for a minimum of 50% with self-reported severe PA and at least 25% who had experienced a life-threatening reaction (defined as requiring intubation or intravenous adrenaline) or used an adrenaline auto-injector (AAI).

Each stage of the study was reviewed and approved by an independent ethics board (APPEAL-1: Freiburger Ethik-Kommission International; APPEAL-2: Western Independent Review Board). Participants were provided with information about the study and checked an informed consent box (APPEAL-1) or gave verbal consent (APPEAL-2) before study participation. For child and teenage participants, caregivers provided verbal consent for their child to participate, and the children and teenagers provided verbal assent, before study participation. The verbal consent procedures were audio recorded following participants' permission to be

recorded. Each interview was recorded and the electronic document was witnessed by a third party [19–21].

## Analysis

APPEAL-1 data were analysed using descriptive statistics. The APPEAL-1 data were analysed for the sample as a whole, with self-reported (eg, adult with PA) and proxy-reported (eg, caregiver reporting impact on their child) data combined. APPEAL-2 demographic and background data were analysed using descriptive statistics. APPEAL-2 interview data were analysed using thematic analysis [23]. This involved a team of analysts coding the qualitative text of the transcripts using a coding frame. APPEAL-2 analysis was assisted by MAXQDA, a qualitative software tool. Saturation, the point at which no new information is obtained from additional qualitative data, was assessed using saturation tables [24]. A conceptual model, which is a visual representation of the themes and relationships between themes as indicated by the data, was developed using the concepts identified during the analysis.

## Results

### Study participants

A total of 284 adults in the UK and Ireland completed the APPEAL-1 survey between 10 November and 11 December 2017, including 97 adults with PA and 187 caregivers (141 parents/46 non-parents). Of the 187 caregivers, 63 were caregivers for adults with PA, 45 for teenagers and 79 for children.

A total of 42 individuals from the UK and Ireland participated in APPEAL-2 (11 adults with PA, 11 teenagers with PA, 8 children with PA [UK only] and 12 caregivers of a child with PA). A summary of the demographic data is shown in **Table 1**. All age groups contained both

**Table 1. Demographics.**

| Characteristic | | APPEAL 1 | | | | APPEAL 2 | | | |
|---|---|---|---|---|---|---|---|---|---|
| | | Adults (self- and proxy- report)[1] | Children: Age, y 0–3 (proxy report) | Children: Age, y 4–12 (proxy report) | Teenagers: Age, y 13–17 (proxy report) | Caregivers[2] | Adults[3] | Children: Age 8–12 y | Teenagers: Age 13–17 y |
| N | | 160 | 10 | 69 | 45 | 12 | 11 | 8 | 11 |
| Age, mean (SD), y | | 33.9 (15.4) | 2.3 (0.7) | 8.4 (2.6) | 15.2 (1.3) | 42.3 (4.6) | 23.4 (4.4) | 9.5 (1.6) | 15.1 (1.2) |
| Gender, n (%) | Male | 63 (39%) | 8 (80%) | 35 (51%) | 31 (69%) | 1 (8%) | 6 (55%) | 6 (75%) | 3 (27%) |
| | Female | 97 (61%) | 2 (20%) | 34 (49%) | 14 (31%) | 11 (92%) | 5 (45%) | 2 (25%) | 8 (73%) |
| Other FA, n (%) | | | | | | | | | |
| | Tree nuts | 76 (48%) | 2 (20%) | 38 (55%) | 22 (49%) | 7 (58%)* | 4 (36%) | 3 (38%) | 5 (45%) |
| | Other FA | 93 (58%) | 7 (70%) | 41 (59%) | 24 (53%) | 2 (17%)* | 5 (45%) | 2 (25%) | 6 (55%) |
| AAI prescribed, n (%) | Yes | 115 (72%) | 7 (70%) | 61 (88%) | 43 (96%) | 9 (75%)* | 10 (91%) | 5 (63%) | 8 (73%) |

FA, food allergy; AAI, adrenaline auto-injector.

*Refers to the FA or AAI prescription of the caregiver's child

[1] Includes 97 adults with PA and 63 caregivers of adults

[2] To children aged 4–17 years

[3] Aged 18–30 years.

male and female participants. Most participants were prescribed an AAI because of their PA (between 63%–96% depending on group). In APPEAL-1, PA was first diagnosed by an allergist for 35% of participants, by other HCPs for 52% of participants, and other/never for 13% of participants. In APPEAL-1, the most commonly reported clinical evaluations used for PA diagnosis were peanut skin prick test (SPT; 66%) and peanut-specific immunoglobulin E (IgE) test (50%), with 34% of respondents reporting both peanut SPT and IgE. In APPEAL-1, 72% of participants were hospitalized and/or used an AAI for their worst reaction to peanut. In APPEAL-2, 64% of participants had used an AAI and 19% had experienced a life-threatening reaction.

## APPEAL-1

The results from the APPEAL-1 survey data showed that individuals with PA and their caregivers report that living with PA means they have restricted choices in various situations. **Fig 1** shows the situations causing the most restrictions for participants: choosing food when eating out (87% reported feeling moderately to extremely restricted), choosing where to eat (82%), special occasions (76%) and when buying food from a shop (71%). Approximately two-thirds of participants also reported they were at least moderately restricted when choosing a holiday destination (68%), traveling on airplanes (68%), when socialising with friends (64%) and when choosing where to buy food (63%).

When asked about the amount of extra planning required in their daily life due to avoiding peanut exposure, two-thirds of participants reported (at least) a moderate amount (68%), and

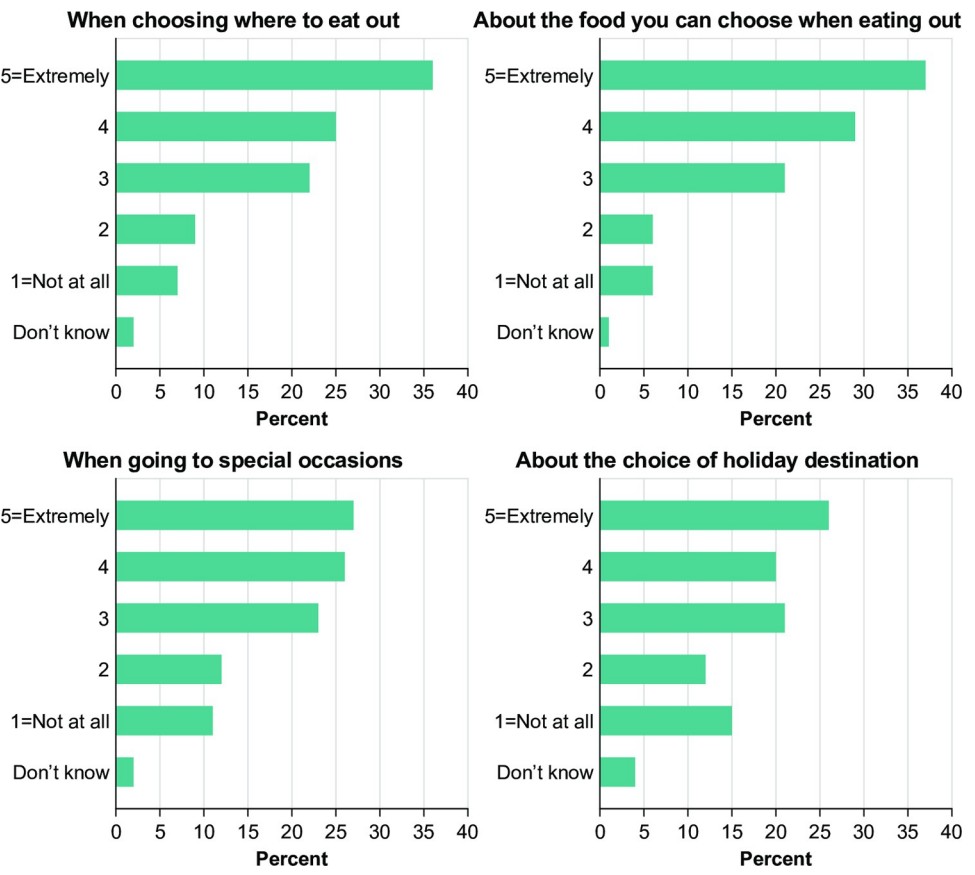

**Fig 1. Restrictions on choice in different situations.**

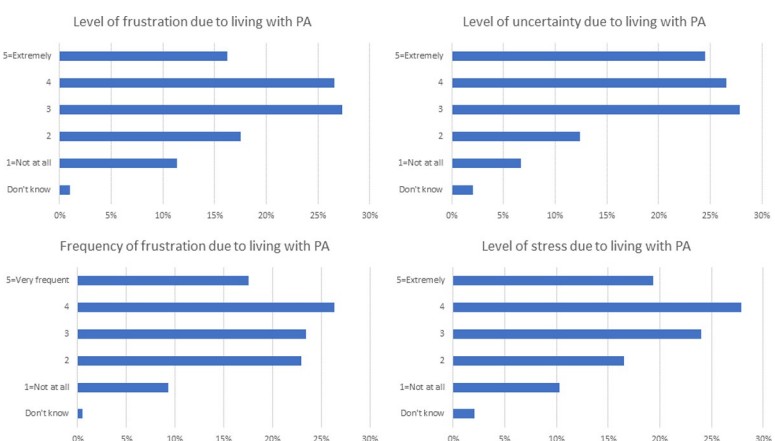

**Fig 2. Levels of frustration, uncertainty and stress of living with PA.**

when asked about extra planning required for special activities (special occasions, holidays) to avoid peanut exposure, 78% reported (at least) a moderate amount.

Almost a fifth of participants (18%) reported that they are very frequently frustrated by the limitations of living with PA, while over two-thirds (70%) reported (at least) a moderate amount of frustration due to living with PA. In addition, as shown in **Fig 2**, many participants reported (at least) moderate levels of uncertainty (79%) and stress (71%) due to living with PA. Two-thirds of participants reported (at least) moderate frequency of feeling tense (65%) and three-quarters reported feeling anxious (75%).

**Fig 3** shows the proportion of participants reporting (at least) a moderate level of worry in several different situations including social occasions where food is involved (86%), and when

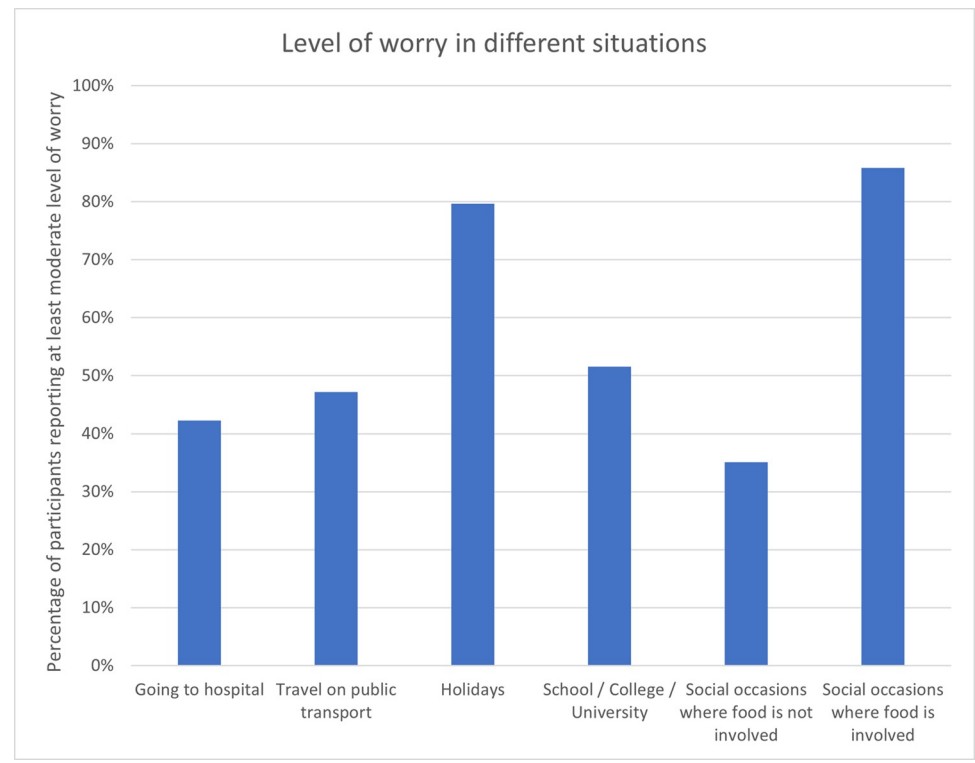

**Fig 3. Level of worry in different situations.**

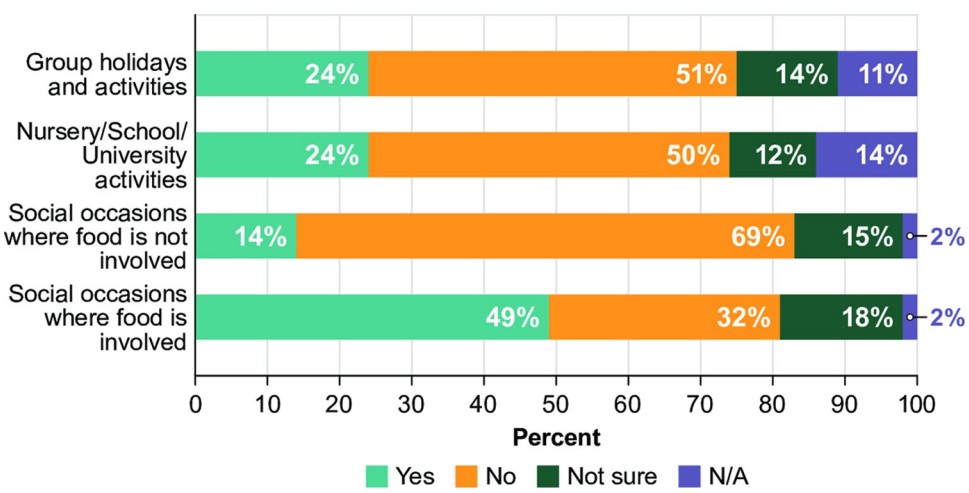

**Fig 4. Excluded from different situations due to PA.**

on holiday (80%). The situations causing the lowest level of worry were the social occasions that did not involve food; however, a third of participants (35%) did report (at least) a moderate level of worry in those situations.

Only a quarter of participants (25%) have never felt different (in a negative way) because of their PA, whereas 28% have been made to feel different quite or very frequently because of their PA. The proportion reporting feeling different because of their PA was greatest in the adolescent age group, where only 16% of the same group reported having never felt different. In addition, over a fifth (22%) of participants have experienced feelings of isolation quite or very frequently because of living with PA. Fifty-two percent of participants reported PA-related bullying, and of these 60% reported that the impact of this is at least moderate. The situation most participants reported being excluded from because of their PA was social occasions involving food (49%); a quarter had also been excluded from group holidays or activities and nursery, school or university activities (**Fig 4**). Most participants (72%) felt that their family had a good awareness and understanding of PA (note: the survey did not provide a specific definition of 'family'), whereas only 51% felt their friends did and only 13% felt other people (excluding family and friends) had a good understanding of PA. Almost all participants (94%) felt that they cope at least moderately well with their PA now compared to when they were diagnosed; however, only 37% of participants now cope 'extremely well.' Almost all (90%) participants who have been prescribed an AAI report that they carry it at least 75% of the time, with two-thirds (65%) carrying it all of the time; if participants forget their AAI, almost all (93%) are at least moderately more anxious than when carrying it.

## APPEAL-2

In interviews, all participants spoke about the need to use coping strategies or behaviours to avoid accidental exposure to peanuts. The qualitative analysis identified three different types of coping strategies (daily monitoring or vigilance, communication and planning) and four main areas of individuals' lives that are impacted by PA (social activities, relationships, emotions and work [adults and caregivers only]). The extent of the impact reported varied substantially between participants, with some reporting many negative consequences of living with PA and others feeling it has minimal impact on their HRQL. The conceptual model (**Fig 5**) illustrates the relationships between the coping strategies and their impact and the moderating role of

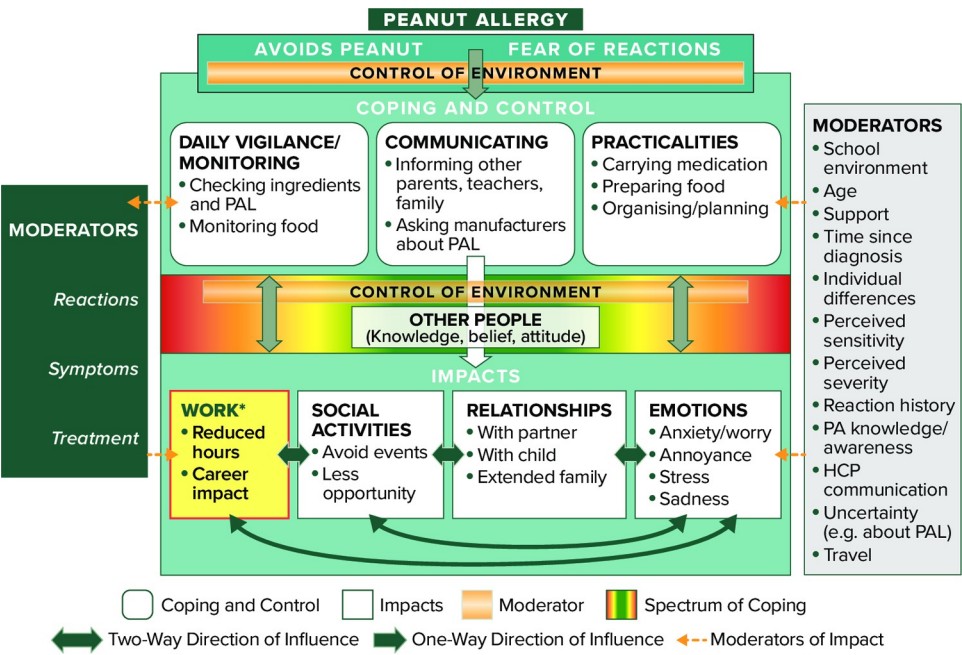

**Fig 5. APPEAL-2 conceptual model: UK and Ireland.**

other factors, such as control, other people and reactions. Descriptions of each of the main elements of the model are provided below, with example quotations provided in **Table 2**.

**Coping strategies.** Strategies relating to daily monitoring and vigilance were reported by all participants; this included checking ingredients, constantly paying attention to what others are eating, staying away from other people if they are eating peanuts and asking other people to wash their hands after eating peanuts. Communication as a strategy involved having to ask about ingredients and having to tell other people about their PA, and for caregivers this included having to inform school staff and other parents. Several child participants were reluctant to tell other people about their PA, either because they were embarrassed or because they wanted to avoid being teased about it. Strategies relating to minimising the chance of a PA reaction included always carrying emergency medication such as an AAI, carefully buying and preparing food, ensuring no peanuts are in their home and planning ahead by checking menus or contacting restaurants.

**Impact on health-related quality of life.** All adult, adolescent and caregiver participants and over half of the children interviewed reported a negative impact of PA on their social activities. This included not going out with friends if they were going to a restaurant (if perceived as unsafe), avoiding parties or events where peanuts might be served, having limited food options when attending social events. Some caregivers also reported supervising their child at parties more often and with a higher degree of vigilance than parents of other children do. Half of caregivers discussed feeling stressed because of their child's PA, particularly related to social occasions.

Most participants reported a negative impact of PA on their relationships, with over half of the caregivers perceiving an adverse impact on their relationship with their partner, mainly due to them feeling that their partner did not pay enough attention to ensuring their child avoided peanuts. Other adverse impacts on participant relationships included being left out of activities with friends and family members not paying attention to their PA. Similar to the APPEAL-1 results, approximately 40% of participants reported that they or their child have

**Table 2. APPEAL-2: Table of example quotes from British and Irish APPEAL-2 participants.**

| | Sample quotes from adults, teenagers, children | Sample quotes from caregivers |
|---|---|---|
| **Daily coping strategies** | | |
| **Daily monitoring/ vigilance** | "If we go to a pub or something, I always make sure that nobody's eating peanuts around me (. . .) just sort of checking what other people are eating constantly, just in case." [Female, Age 23, UK] | "I'm a bit scared about bowls, to be honest. I tend to, if it's gone through the dishwasher, [. . .] whenever I'm giving him a bowl, I'm always rinsing it with hot water before I give it to him. I don't know whether I'm becoming overly obsessive with it as well. But I'm just. . . erring on the side of caution." [Female caregiver of boy aged 9 y, UK] |
| **Communicating** | "I think the most time and effort comes from like telling people about the allergies, like restaurants and things. Because. . . It's always the same spiel every time. I've got an allergy, can you make sure, this, that and the other" [Male, Age 19, UK] | "So, I feel almost like we've had to educate family, especially my mum and dad, and it's like, 'Oh, he's allergic to peanuts.' 'Yes, and it can kill him,' do you know what I mean? And I feel like I have to follow that up with that. This allergy could kill him, and I always feel that I have to stress that. Going round to friends' houses and things like that." [Female caregiver of boy aged 8 y, UK] |
| **Practicalities** | "Well, yes obviously because I can't eat things with peanuts in. But when I go to restaurants and stuff, that is the most annoying because you have to check the menu before, and maybe go on the website and check how they deal with peanut allergies and stuff like that." [Female, Age 16, UK] | "I cook a lot now. [. . .] Way more than I used to. So, I cook a lot more because–he's a good eater–I want to ensure that he has enough variety while minimising his risk, I suppose." [Female caregiver of boy aged 8 y, UK] |
| **HRQL impacts** | | |
| **Social/ school activities** | "I'm sometimes trying to get out of going for a meal, or going to special occasion parties and stuff, just because I don't know what's going to be on the buffet. Or, I just won't eat at all, which is wrong really." [Male, Age 25, UK] | "Social occasions are stress. [Child] going to anybody's house. I–not discourage–I'm mindful, I would say. I would not shy away from, that's the wrong expression, I would be reluctant to push him really. I'm happier for him to be at home, so it's something that I don't have to think about." [Female caregiver of boy aged 8 y, UK] |
| **Relationships** | "I just feel bad, like, when I go out for food with my boyfriend, and things like that, I just feel bad that I'm always the one choosing the restaurant, and things like that. It probably doesn't bother him as much as it bothers me, but I just feel bad, and, like, same as when I'm shopping with people, I feel like they're just having to wait around for me to check everything." [Female, Age 16, UK] | "It can cause conflict [with husband]. [. . .] I'm a control freak. [. . .] So it does have a negative effect." [Female caregiver of boy aged 9 y, UK] |
| **Emotions** | "It is scary, knowing that. Because I don't know what people have eaten, I don't sit there and watch what everyone's eaten around me. So, I could go past, touch something that they've had nuts on, and it could react with me. It's scary to think, because I could touch anything and react badly to it." [Female, Age 16, UK] | "And now he's 11, a lot more independent, I'm starting to feel anxious about what this next bit's going to bring, because I'm not always there, hovering over him, checking things for him, so they are the biggest issues, more the worrying and the making sure people know" [Female caregiver of a boy aged 11 y, UK] |
| **Work** | "Yeah, if I have a reaction that whole day is gone. And maybe even the next day, I don't feel that well after having adrenaline. So every time I've had a reaction it has affected my performance, in work or college." [Female, Age 28, Ireland] | "There's this whole plethora of stuff going that has to be managed, to try and maintain some kind of status quo. I was a Deputy in [School]. So, it was a conversation where something has to give, so, now I'm a part-time teacher." [Female caregiver of boy aged 8 y, UK] |

experienced bullying or teasing because of their PA, including half of the children and over half of the teenagers interviewed. The incidence included classmates waving peanuts around near them and making jokes about their PA.

Almost all participants reported an adverse emotional impact because of PA, primarily relating to anxiety or worry about experiencing a reaction. Many caregivers also worried about their child's future when the caregiver would have less control over their food, behaviour and environment. There were two main stress points, either when their child reached adolescence and spent more time with friends, or when they left home as an adult and the caregiver would have less control over their food and environment. Most adolescent and adult participants felt annoyed or frustrated about the vigilance required due to PA, including having to always check ingredients or having a lack of food options. Children most commonly reported feeling sad because they could not eat things other children could eat.

Some adult and caregiver participants reported a negative impact of PA on their work or career. A small number of adults had experienced a reaction to peanut whilst at work, which

**Table 3. Case studies outlining a participant reporting minimal, moderate or severe impact from APPEAL-2.**

| | Minimal impact | Moderate impact | Severe impact |
|---|---|---|---|
| **Demographics** | Female, Age 12 y | Male, Age 9 y | Female, Age 16 y |
| **AAI/confidence/ control** | Yes/confident/good amount | Yes/somewhat/some | Yes/somewhat/good |
| **Reaction history** | Two reactions, had to use AAI for one | Has had three reactions. Most recent reaction was two years ago, treated with antihistamines. | Three reactions, more severe as she's gotten older. Had to use AAI and go to hospital for most recent reaction (2 years ago). |
| **Main impacts** | Always having to check labels is the hardest aspect of PA. | Feels worried because it can cause bad reactions and death, usually only worried when he has a reaction. | Reaction at school: missed an afternoon of lessons. |
| | She does not tell people about her PA unless she has to disclose. | He has to be wary having dinner outside home, but not at school as they know about his PA. | She does not want to say to new friends that they cannot eat nuts around her. |
| | She feels disappointed when she cannot eat something. | He feels a bit sad that he cannot try some foods. | She has fear related to her AAI (expiring, who would use it). |
| | | | Her PA impacted her choice of college (closer to home). |
| | | | Situations such as planes, buses and eating out are stressful. |
| | | | She feels on edge and would not want to eat if she does not have her AAI. |
| | | | Children in school used to throw peanuts around near her knowing she was allergic to them. |

had meant they had to go home. Other impacts on work included having to avoid colleagues who are eating peanuts or having to request that colleagues do not eat peanuts. One caregiver had reduced their working hours to part-time and taken a job with less responsibility to take their child to allergy appointments (as well as other medical appointments). Other caregivers reported having to take time off work for appointments or when their child had experienced a reaction.

The analysis also identified two important moderators that participants discussed having a positive or negative impact on their HRQL: level of perceived control (over environment and food) and level of perceived awareness (other people's knowledge of and attitude towards PA). Each of these moderators had an impact on the extent to which participants' social activities, emotions, relationships and work were affected (positively or negatively).

Table 3 shows three example case studies from the APPEAL-2 sample. The profiles summarise the demographics, self-reported severity, level of confidence in managing PA and control of PA, AAI prescription, reaction history, and the main impacts reported in the interview for a participant who reported minimal impact, one who reported moderate impacts, and one who described a severe impact of PA on their HRQL. These examples demonstrate that people's reported levels of control and confidence with PA may not correlate with the impact PA has on their lives.

## Discussion

The APPEAL study was a large quantitative and qualitative study assessing the psychosocial burden of PA in eight European countries, with the focus of the current article on the experience of participants in the UK and Ireland. The results detailed in this paper extend our understanding of the substantial burden of PA in the daily life of individuals with PA and their caregivers, compared with other quality-of-life surveys of PA patients and caregivers in other countries, including the US, Canada, and Switzerland, by highlighting that it is situation specific and because this study also examined coping strategies [25–28]. The survey phase of the

study showed the high proportion of people with PA and their caregivers reporting negative emotions such as frustration, stress, worry and uncertainty due to PA. The qualitative data highlight the different coping strategies used and the wide variation in the level of impact, with some participants reporting high levels of burden while others with the same perceived severity of PA report minimal burden.

The results from the full APPEAL study showed the psychosocial burden of PA in Europe [20, 21]; this study provides an in-depth view of the results specific to the UK and Ireland and adds to previous research conducted in these countries. Recent research found that UK caregivers of children with PA experience greater anxiety than the general population [15]; this is supported by the APPEAL-1 results showing the many situations causing worry for individuals with PA and their caregivers and the APPEAL-2 data in which anxiety or worry related to PA was reported by all caregivers and adults with PA, as well as almost all teenagers and several children.

Previous research on coping mechanisms used in food allergy demonstrated different strategies used by individuals at different development stages [29]. Strategies identified, including avoidance (for example avoiding risky places), minimisation (rejection of the food-allergic identity) and adaptive strategies (positive behaviours, such as openly telling people about their food allergy), were also seen in the qualitative results in the current study as ways that individuals in each age group coped with their PA. Much of the data indicate a poor understanding of risk of accidental reactions, suggesting that access to expert advice is needed for the most part.

The use of both quantitative and qualitative methods means that the interview data can add depth to the findings from the survey. For example, the survey data showed that most participants report an impact of PA on their daily activities; the qualitative data provide specific ways in which individuals' daily lives are impacted by the strategies they use to avoid peanuts (monitoring, vigilance, communication). The survey data also show the various psychosocial impacts of PA that were commonly reported, such as feeling frustrated, worried, uncertain or stressed; similar impacts were reported spontaneously in the qualitative data, therefore providing support for the quantitative findings and corroborating the links between views in the conceptual model (eg, the survey data showed certain social situations cause worry, supporting the link between emotions and social activities). Both the survey and qualitative data showed that being bullied owing to PA is fairly common (52% of survey participants and 40% of interview participants). This, along with the low proportion who felt that other people have a good understanding of PA, suggests that more public awareness and understanding of PA is required in the UK and Ireland. The conceptual model also showed the important role other people can play in reducing the negative impact on individuals with PA.

The survey data show that despite a high proportion of participants reporting each of the psychosocial impacts, almost all participants report that they cope well with their PA. The qualitative data highlighted in the case studies show that although someone reports 'good control' and confidence managing their PA, obtaining this level of control and confidence may be causing a decrement to their HRQL; therefore, simply measuring control of PA or confidence in managing PA should not be considered sufficient in determining that the individual does not need additional support. The qualitative data highlight the issue that some people with PA manage well with minimal HRQL impact, however, for some there is an unmet need for a treatment that reduces the impact of PA on their life. Future research could use the conceptual model to develop hypotheses to explore using the quantitative data, for example using structured equation modelling.

Some limitations should be considered when interpreting the results of this study. Although the protocols allowed subjective reporting of PA diagnosis and severity, the majority of respondents in the APPEAL-1 sample indicated that the first PA diagnosis was made by an allergist

and was supported using an objective measure, such as a peanut-specific IgE test, peanut SPT, or both. The APPEAL-1 survey did not include any reports directly from children or teenagers with PA, therefore the views of children and teenagers were not captured quantitatively. As previous research in food allergy has found that parents reported significantly better HRQL for their child than the children themselves reported, there is a possibility that the results of APPEAL-1 underestimate the psychosocial burden experienced by children and teenagers with PA. Although each age group in both samples contained both male and female participants, there was only one male caregiver in APPEAL-2, therefore the experience of male caregivers may be under-represented in the qualitative results. Finally, for practical reasons due to the timing of the two phases of the study, it was not possible to capture qualitative and quantitative data from the same participants; however, the finding that the two samples independently reported similar impacts adds validity to the results.

Although findings may reflect a response bias as well as differences in the recruiting process (eg, age of respondents, recruitment through patient advocacy groups vs the professional recruitment service), there were some differences in the patterns of responses from the UK and Ireland compared with the other European countries surveyed [19, 20]. Reported restrictions on activities and the proportion of respondents found to have a "high level" of uncertainty and a "high level" of stress in Ireland was almost twice that of other countries in Europe [19, 20], whereas rates of bullying associated with PA was much higher in the UK compared with other parts of Europe. Additionally, differences were identified through the qualitative interviews—the aspect of PA avoidance involving hygiene (eg, handwashing and asking others to wash their hands) was reported by participants in the UK and Ireland and few other countries in Europe. Finally, the APPEAL-1 survey also showed higher proportions of Irish participants than UK participants reporting a moderate impact on several questions, including level of uncertainty of living with PA (62% vs 47%), level of stress of living with PA (55% vs 45%) and level of frustration of living with PA (78% vs 67%). These differences warrant further examination.

## Conclusions

This large survey and interview study highlights the psychosocial burden of PA in the UK and Ireland for adults, teenagers, children and caregivers. The results demonstrate the wide variation in level of impact of PA and the unmet need for those individuals who experience a substantial burden from living with PA.

## Acknowledgments

The authors would like to express their great appreciation and gratitude to the APPEAL investigators and the patient advocacy groups, physicians and academicians involved in the APPEAL study for their support and valuable input into the study design and writing of the manuscript. Brainsell provided analytical support for this study.

## Author Contributions

**Conceptualization:** Katy Gallop, Ram Patel, Robert Ryan, Andrea Vereda, Sarah Acaster, Audrey DunnGalvin.

**Investigation:** Katy Gallop, Ram Patel, Robert Ryan, Andrea Vereda, Sarah Acaster, Audrey DunnGalvin.

**Methodology:** Katy Gallop, Ram Patel, Andrea Vereda, Sarah Acaster, Audrey DunnGalvin.

**Project administration:** Katy Gallop, Ram Patel, Andrea Vereda, Sarah Acaster, Audrey DunnGalvin.

**Resources:** Robert Ryan, Sarah Acaster, Audrey DunnGalvin.

**Supervision:** Marina Tsoumani, Lynne Regent, Amena Warner, Katy Gallop, Ram Patel, Robert Ryan, Andrea Vereda, Sarah Acaster, Audrey DunnGalvin, Aideen Byrne.

**Writing – original draft:** Marina Tsoumani, Lynne Regent, Amena Warner, Katy Gallop, Ram Patel, Robert Ryan, Andrea Vereda, Sarah Acaster, Audrey DunnGalvin, Aideen Byrne.

**Writing – review & editing:** Marina Tsoumani, Lynne Regent, Amena Warner, Katy Gallop, Ram Patel, Robert Ryan, Andrea Vereda, Sarah Acaster, Audrey DunnGalvin, Aideen Byrne.

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
