## [Decision Letter · Decision Letter 0]

15 Apr 2021

PONE-D-20-40094

Allergy to Peanuts imPacting Emotions And Life (APPEAL): the impact of peanut allergy on children, teenagers, adults and caregivers in the UK and Ireland

PLOS ONE

Dear Dr. Ryan,

Thank you for submitting your manuscript to PLOS ONE. After careful consideration, we feel that it has merit but does not fully meet PLOS ONE’s publication criteria as it currently stands. Therefore, we invite you to submit a revised version of the manuscript that addresses the points raised during the review process.

We look forward to receiving your revised manuscript.

Kind regards,

Janhavi Ajit Vaingankar

Academic Editor

PLOS ONE

Journal Requirements:

2. Please provide additional details regarding participant consent.

In the ethics statement in the Methods and online submission information, please ensure that you have specified what type you obtained (for instance, written or verbal, and if verbal, how it was documented and witnessed).

If your study included minors, state whether you obtained consent from parents or guardians.

If the need for consent was waived by the ethics committee, please include this information.

4. Thank you for stating the following in the Financial Disclosure section:

'This study was sponsored by Aimmune Therapeutics, a Nestle Health Science company. Brainsell provided analytical support for this study. Editorial assistance and submission support were provided by The Curry Rockefeller Group, LLC (Tarrytown, NY); both were funded by Aimmune Therapeutics, a Nestle Health Science company.'

We note that one or more of the authors have an affiliation to the commercial funders of this research study: Aimmune Therapeutics.

We also note that one or more of the authors are employed by other commercial companies: Acaster Lloyd Consulting, Brainsell Ltd. and Allergy UK.

a. Please provide an amended Funding Statement declaring these commercial affiliations, as well as a statement regarding the Role of Funders in your study. If the funding organization did not play a role in the study design, data collection and analysis, decision to publish, or preparation of the manuscript and only provided financial support in the form of authors' salaries and/or research materials, please review your statements relating to the author contributions, and ensure you have specifically and accurately indicated the role(s) that these authors had in your study. You can update author roles in the Author Contributions section of the online submission form.

b. Please also provide an updated Competing Interests Statement declaring these commercial affiliations along with any other relevant declarations relating to employment, consultancy, patents, products in development, or marketed products, etc.  

Within your Competing Interests Statement, please confirm that these commercial affiliations do not alter your adherence to all PLOS ONE policies on sharing data and materials by including the following statement: "This does not alter our adherence to  PLOS ONE policies on sharing data and materials.” (as detailed online in our guide for authors http://journals.plos.org/plosone/s/competing-interests). If this adherence statement is not accurate and  there are restrictions on sharing of data and/or materials, please state these. Please note that we cannot proceed with consideration of your article until this information has been declared.

Reviewers' comments:

Reviewer's Responses to Questions

**Comments to the Author**

1. Is the manuscript technically sound, and do the data support the conclusions?

Reviewer #1: Yes

Reviewer #2: Yes

2. Has the statistical analysis been performed appropriately and rigorously? 

Reviewer #1: I Don't Know

Reviewer #2: I Don't Know

3. Have the authors made all data underlying the findings in their manuscript fully available?

Reviewer #1: No

Reviewer #2: Yes

4. Is the manuscript presented in an intelligible fashion and written in standard English?

Reviewer #1: Yes

Reviewer #2: Yes

5. Review Comments to the Author

Reviewer #1: This is another publication generated by APPEAL-1 and APPEAL-2

The self-reported nature of the peanut allergy, the lack of previously published self-reported medical data, and the fact that these respondents' data points were already analyzed (with the exception of 11 adults apparently not included in the previous APPEAL-2 publication), raises expectations that another manuscript out of the database should be able to stand alone and enhance understanding compared to the previous multiple publications.

I would urge the authors to include more information that may have been included in the questionnaires used by the recruiters but not published. In the APPEAL-2 group, how did you define moderate or severe PA? In parentheses on p 5, line 120, it says ".....severe PA (PA diagnosed by a medical doctor)." I would refute the statement that a "medical doctor diagnosis" of PA is a definition of "severe PA." A severe peanut allergy is based on clinical symptoms combined with proof of sIgE. This raises concerns that it is possible that these children and young adults are living in fear and yet do not have a proven food allergy. We all know this happens and much of our job as allergists is performing food challenges to "clear" food allergy in those who have been diagnosed by a GP doing a "food panel". Hives are common and may not be a food allergy, especially if someone has a grass or birch allergy and the GP did a Peanut ImmunoCAP rather than components.

For example, how many self-reported that they had confirmatory sIgE testing, skin prick testing or oral food challenge? How many had ED visits, hospitalizations, intubations, or had used their EAIs? I would like to know more specifically how you analyzed the data from those with more severe allergy compared to milder allergy, noting that only 5 of the 8 children were prescribed EIAs, and only 8 of 11 teens were likewise prescribed EIAs and yet the enlistment criteria were self-reported moderate or severe PA. The authors state that recruitment aimed for minimum of 50% with self-reported severe PA (but again the definition could have included itchy mouth if it was "diagnosed by a doctor" per the criteria outlined) and at least 25% who had used their EIAs or were intubated or had epinephrine by vein. The paper does not show whether these recruitment goals were reached.

Also, this study included 11 adults who were reported as part of APPEAL-2, but I don't' see adults listed in the table for APPEAL-2 in DunnGalvin et al. Does this mean there is another publication collating adults not reported as part of APPEAL-2 in another publication for pan-Europe that is in the review process at another journal? I checked the supplementary information in the original publication - there is NOT AN INTAKE FOR ADULTS WITH PA, NOR AN INTERVIEW SCRIPT.

Also, the intake form does not ask explicit medical questions and just asks when PA was diagnosed by a doctor or NURSE.

I don't see substantial differences in the model generated compared with what was already published in APPEAL-2 DunnGalvin. How does Figure 5 in this paper contribute something new? Also, I cannot tell how you did the statistical analysis for this Figure - it talks about "CHILD" but you have included TEENS, ADULTS, AND CAREGIVERS in your methods section. Are all 4 groups in this one conceptual model? There was no figure legend included in the dowloaded submission that I accessed.

Other Specific Comments

1. Abstract: Would insert "self-reported" prior to "peanut allergy" in line 25

2. Abstract: line 29: would point out that this conceptual model is unique to the UK and Ireland. Also, would include a line about results reported on bullying in the abstract and include this in key words.

3. Introduction: insert "the" between "highlights" and "burden" in line 78

4. Table 1: APPEAL-1. How many adults were self, and how many proxy

5. Table 1: age - in the first column , authors left off "(mean)"

6. Table 1: APPEAL-2 - please explain the asterisk after the number of Caregivers reporting on 7 with tree nut allergy. In the footnote, it says "Child's FA/AAI prescription" Does this mean some children had tree nut allergy as the reason for the AAI prescription but not for peanut allergy or in addition? And does the lack of asterisk for "other" mean that the other food allergies did not have Epi prescribed?

7. Results: APPEAL-1: p 10, Were the results homogenous statistically between age and being a proxy reporter or not? Otherwise, why were the data analyzed for the sample as a whole? Please state why the data were not analyzed separately and compared statistically between groups, if if this was done but not significant, state it and which statistics were used in the Methods section.

8. Results: Fig 3, text p 11. The authors state that only a quarter of participants had never felt different. This would be really interesting to have AGE data regarding.

9. Results Fig 4, text, p 11. Only 72% felt family had a good awareness. How was family defined? Extended or close circle. If close, this figure is shockingly low. If extended, that would explain it.

10. Fig 5 - again, how was this modeled and why is it focused on "child." - is the model for adults different? (see above in general comments)

11. Table 2. The authors have repeated quotes that were already published in the APPEAL-2 paper. Recommend publishing new quotes.

12. Discussion: the discussion of limitations on p 11 needs to explicitly state the problems with this being self-reported peanut allergy. What recruitment company was used (should be in Methods)?

Reviewer #2: I would like to thank you for the opportunity to review this manuscript.

In the introduction, all 8 countries participating in the APPEAL study should be specified. Further, the reason that led the authors to search for differences in the UK and Ireland populations as compared to the others (different background, different culture, different kinds of food, etc) should be specified.

Fig.2 would be easier to understand if displayed as a column.

In the Discussion session, I would suggest emphasizing differences between the UK and Ireland participants as well as those from the other countries that took part to the study. Moreover, I would suggest comparing the results of this manuscript with results from manuscripts from other countries (eg, Nowak-Wegrzyn A, et al.The Peanut Allergy Burden Study: Impact on the quality of life of patients and caregivers. World Allergy Organ J. 2021 Feb 15;14(2):100512).

In conclusion, I think this is manuscript is well written and details the multiple problems a peanut- allergic patient may present during own life.

6. PLOS authors have the option to publish the peer review history of their article (what does this mean?). If published, this will include your full peer review and any attached files.

Reviewer #1: No

Reviewer #2: No

---

## [Author Response · Author response to Decision Letter 0]

30 Jun 2021

Date: June 28, 2021

Manuscript Number: PONE-D-20-40094

Original Manuscript Title: Allergy to Peanuts imPacting Emotions And Life (APPEAL): the impact of peanut allergy on children, teenagers, adults and caregivers in the UK and Ireland

Name of the Corresponding Author: Audrey DunnGalvin, MD

Email Address of the Corresponding Author: A.DunnGalvin@ucc.com

Dear Dr Vaingankar,

Thank you for reviewing our manuscript and considering it for publication in PLOS ONE. The following is a point-by-point response to the reviewer comments delivered in your letter dated April 15, 2021. 

Major changes and additions to the revised manuscript:

1. Added information on first diagnosis and clinical evaluations to the Results

2. Amended Figure 5

3. Added sentence regarding bullying to the Abstract and added “bullying” to keywords

4. Revised Table 1 footnotes

5. Removed previously published quotes from Table 2 and added new quotes

6. Added limitations of self-reported PA diagnosis and severity to Discussion

7. Revised Figure 2 using bar graphs

8. Added notes regarding consistency of the results with those from other countries (including 4 additional references) as well as differences between Ireland and UK to the Discussion

9. Clarified how consent was obtained in the Methods

10. Updated Author Contributions section, Funding statement, and Competing Interests Statement

Response to Reviewer #1:

This is another publication generated by APPEAL-1 and APPEAL-2 The self-reported nature of the peanut allergy, the lack of previously published self-reported medical data, and the fact that these respondents' data points were already analyzed (with the exception of 11 adults apparently not included in the previous APPEAL-2 publication), raises expectations that another manuscript out of the database should be able to stand alone and enhance understanding compared to the previous multiple publications.

MAJOR COMMENTS:

1. I would urge the authors to include more information that may have been included in the questionnaires used by the recruiters but not published. In the APPEAL-2 group, how did you define moderate or severe PA? In parentheses on p 5, line 120, it says “.....severe PA (PA diagnosed by a medical doctor).” I would refute the statement that a “medical doctor diagnosis” of PA is a definition of “severe PA.” A severe peanut allergy is based on clinical symptoms combined with proof of sIgE. This raises concerns that it is possible that these children and young adults are living in fear and yet do not have a proven food allergy. We all know this happens and much of our job as allergists is performing food challenges to “clear” food allergy in those who have been diagnosed by a GP doing a “food panel”. Hives are common and may not be a food allergy, especially if someone has a grass or birch allergy and the GP did a Peanut ImmunoCAP rather than components. For example, how many self-reported that they had confirmatory sIgE testing, skin prick testing or oral food challenge? How many had ED visits, hospitalizations, intubations, or had used their EAIs?

Response: We appreciate the reviewer’s comment. Details regarding the first diagnosis and method of diagnosis broken out by country have been previously published for APPEAL-1 [1]. We have added a summary of the UK and Ireland data to the Results (Study participants, lines 162–167) section as follows: “In APPEAL-1, PA was first diagnosed by an allergist for 35% of participants, by other HCPs for 52% of participants, and other/never for 13% of participants. In APPEAL-1, the most commonly reported clinical evaluations used for PA diagnosis were peanut skin prick test (SPT; 66%) and peanut-specific immunoglobulin E (IgE) test (50%), with 34% of respondents reporting both peanut SPT and IgE. In APPEAL-1, 72% of participants were hospitalized and/or used an AAI for their worst reaction to peanut. In APPEAL-2, 64% of participants had used an AAI and 19% had experienced a life-threatening reaction.” Furthermore, participants were excluded from APPEAL-2 if they had never experienced a reaction to peanut in their day-to-day life.

2. I would like to know more specifically how you analyzed the data from those with more severe allergy compared to milder allergy, noting that only 5 of the 8 children were prescribed EIAs, and only 8 of 11 teens were likewise prescribed EIAs and yet the enlistment criteria were self-reported moderate or severe PA. The authors state that recruitment aimed for minimum of 50% with self-reported severe PA (but again the definition could have included itchy mouth if it was “diagnosed by a doctor” per the criteria outlined) and at least 25% who had used their EIAs or were intubated or had epinephrine by vein. The paper does not show whether these recruitment goals were reached.

Response: The recruitment goals were reached. The severity level was self-reported by participants and was their own perception of how severe their/their child’s PA was. No further definitions of the severity categories were provided. This classification was chosen based on input from clinicians who advised that because there is no objective definition of severity, self-report was appropriate. However, the PA severity of the sample was enriched by the quotas regarding life-threatening events and AAI use. We did not conduct analysis comparing participants with different self-reported severity levels or any other metric of severity.

3. Also, this study included 11 adults who were reported as part of APPEAL-2, but I don’t see adults listed in the table for APPEAL-2 in DunnGalvin et al. Does this mean there is another publication collating adults not reported as part of APPEAL-2 in another publication for pan-Europe that is in the review process at another journal? I checked the supplementary information in the original publication - there is NOT AN INTAKE FOR ADULTS WITH PA, NOR AN INTERVIEW SCRIPT.

Response: The adult participants were not included in the DunnGalvin et al APPEAL-2 publication because it was decided that that article would focus on the experience of children/caregivers. There is not currently another publication planned to report the adult data; therefore, it was incorporated into this manuscript. There was a separate interview script for adults, which was not included in the previous supplementary materials because the data were not reported there. 

3. Also, the intake form does not ask explicit medical questions and just asks when PA was diagnosed by a doctor or NURSE. I don't see substantial differences in the model generated compared with what was already published in APPEAL-2 DunnGalvin. How does Figure 5 in this paper contribute something new? Also, I cannot tell how you did the statistical analysis for this Figure - it talks about “CHILD” but you have included TEENS, ADULTS, AND CAREGIVERS in your methods section. Are all 4 groups in this one conceptual model? There was no figure legend included in the downloaded submission that I accessed.

Response: The model includes all 4 groups in one model because the overall concepts were the same for each group; the work concept applies to both caregivers and adults with PA. The model has been amended slightly. The model was developed from the qualitative data; therefore, no statistical analysis was conducted to develop the model. Indeed we have suggested in the Discussion that future research could use quantitative data to explore the relationships in the model as it is currently based only on qualitative data. 

OTHER SPECIFIC COMMENTS:

1. Abstract: Would insert “self-reported” prior to “peanut allergy” in line 25

Response: Added as suggested.

2.Abstract: line 29: would point out that this conceptual model is unique to the UK and Ireland. Also, would include a line about results reported on bullying in the abstract and include this in key words.

Response: We have revised the sentence in the Abstract (lines 30–31) to read: “A conceptual model specific to UK and Ireland was developed…” Also, we have added the following sentence to the Abstract (lines 35–36): “Fifty-two percent of survey participants and 40% of interview participants reported being bullied because of PA.” In addition, we have added “bullying” to the keywords.

3. Introduction: insert “the” between “highlights” and “burden” in line 78

Response: Added as suggested.

4. Table 1: APPEAL-1. How many adults were self, and how many proxy

Response: In the Results (Study participants) section, we have indicated that in APPEAL-1, the participants included 97 adults with PA and 63 caregivers for adults. For clarity, we have added a footnote to Table 1 to indicate the numbers of self- and proxy reports for adult participants.

5. Table 1: age - in the first column , authors left off “(mean)”

Response: This has been corrected.

6. Table 1: APPEAL-2 - please explain the asterisk after the number of Caregivers reporting on 7 with tree nut allergy. In the footnote, it says “Child's FA/AAI prescription” Does this mean some children had tree nut allergy as the reason for the AAI prescription but not for peanut allergy or in addition? And does the lack of asterisk for "other" mean that the other food allergies did not have Epi prescribed?

Response: The asterisk was inadvertently left off the “Other FA” category for Caregivers and has been added. The footnote for the asterisk has been revised for clarification as follows: “*Refers to the FA or AAI prescription of the caregiver’s child.”

7. Results: APPEAL-1: p 10, Were the results homogenous statistically between age and being a proxy reporter or not? Otherwise, why were the data analyzed for the sample as a whole? Please state why the data were not analyzed separately and compared statistically between groups, if this was done but not significant, state it and which statistics were used in the Methods section.

Response: The study was not designed or powered for statistical analysis. 

8. Results: Fig 3, text p 11. The authors state that only a quarter of participants had never felt different. This would be really interesting to have AGE data regarding.

Response: Thank you for this suggestion. We have amended the Results (APPEAL-1 section, lines 240–241) with some detail on age differences related to feeling different because of PA as follows: “The proportion reporting feeling different because of their PA was greatest in the adolescent age group, where only 16% of the same group reported having never felt different.”

9. Results Fig 4, text, p 11. Only 72% felt family had a good awareness. How was family defined? Extended or close circle. If close, this figure is shockingly low. If extended, that would explain it.

Response: A specific definition for family was not provided to the respondents. It is therefore plausible that the respondents’ answers included extended family. We have clarified this in the Results (APPEAL-1 section, line 212–213) as follows: “Most participants (72%) felt that their family had a good awareness and understanding of PA (note: the survey did not provide a specific definition of ‘family’),…” 

10. Fig 5 - again, how was this modeled and why is it focused on “child.” - is the model for adults different? (see above in general comments)

Response: As described in the response to Major Comment #3 above, the model was developed based on the qualitative data and is an illustration of the areas of HRQL reported to be impacted by PA. The model also includes adults with PA, who reported an impact on their work (along with caregivers); the other concepts apply to both adults and children. We have amended the model slightly to reflect this. 

11. Table 2. The authors have repeated quotes that were already published in the APPEAL-2 paper. Recommend publishing new quotes.

Response: As suggested, in Table 2 we have removed quotes that have been published previously and added new quotes.

12. Discussion: the discussion of limitations on p 11 needs to explicitly state the problems with this being self-reported peanut allergy. 

Response: We have added the following sentence to the second-to-last paragraph in the Discussion lines 343–346: “Although the protocols allowed subjective reporting of PA diagnosis and severity, the majority of respondents in the APPEAL-1 sample indicated that the first PA diagnosis was made by an allergist and was supported using an objective measure, such as a peanut-specific IgE test, peanut SPT, or both.” 

13. What recruitment company was used (should be in Methods)?

Response: Please refer to the Participants section of the Methods for the description of how recruitment was carried out. 

Response to Reviewer #2:

I would like to thank you for the opportunity to review this manuscript.

1. In the introduction, all 8 countries participating in the APPEAL study should be specified. Further, the reason that led the authors to search for differences in the UK and Ireland populations as compared to the others (different background, different culture, different kinds of food, etc) should be specified.

Response: We have added all 8 countries that participated in the APPEAL study to the last paragraph of the Introduction (line 79) as follows: “…conducted across eight European countries (Germany, UK, France, Spain, Denmark, Ireland, Italy and the Netherlands).” 

The reason for reporting differences in the UK and Ireland as compared to the other countries has been added to the last paragraph of the Introduction (lines 80–8) as follows: “…however, this article focuses on the results collected from participants in the UK and Ireland only because they have similar cultures, food habits, and healthcare systems.”

2. Fig.2 would be easier to understand if displayed as a column.

Response: We have revised Fig. 2 using bar graphs to display the data as suggested.

3. In the Discussion session, I would suggest emphasizing differences between the UK and Ireland participants as well as those from the other countries that took part to the study. Moreover, I would suggest comparing the results of this manuscript with results from manuscripts from other countries (eg, Nowak-Wegrzyn A, et al.The Peanut Allergy Burden Study: Impact on the quality of life of patients and caregivers. World Allergy Organ J. 2021 Feb 15;14(2):100512).

Response: Please note that differences in study results between this analysis and other European countries that took part in the APPEAL studies have already been discussed in the last paragraph of the Discussion, lines 357-366. The same areas of impact were reported in the qualitative data in the UK and Ireland. We have added the following sentence to the last paragraph of the Discussion: “Finally, the APPEAL-1 survey also showed higher proportions of Irish participants than UK participants reporting a moderate impact on several questions, including level or uncertainty of living with PA (62% vs. 47%), level of stress of living with PA (55% vs. 45%), and level of frustration of living with PA (78% vs. 67%).”

Regarding comparisons with the results of studies from other countries, we have revised the following sentence in the first paragraph of the Discussion lines 297-300 as follows: “The results detailed in this paper extend our understanding of the substantial burden of PA in the daily life of individuals with PA and their caregivers, compared with with other quality-of-life surveys of PA patients and caregivers in other countries, including the US, Canada, and Switzerland, by highlighting that it is situation specific and because this study also examined coping strategies [25-28].”

4. In conclusion, I think this is manuscript is well written and details the multiple problems a peanut-allergic patient may present during own life.

Response: Thank you for your review.

Journal Requirements:

Response: We have made sure that our manuscript meets PLOS ONE’S style requirements.

Response: We have revised the sentence in the Methods (Participants) section lines 133–137 regarding consent to state the following: “Participants were provided with information about the study and checked an informed consent box (APPEAL-1) or gave informed verbal consent (APPEAL-2) before study participation. For child and teenage participants, caregivers provided verbal consent for their child to participate, and the children and teenagers provided verbal assent, before study participation. The verbal consent procedures were audio recorded following participants’ permission to be recorded.” 

3. We note that you have indicated that data from this study are available upon request. PLOS only allows data to be available upon request if there are legal or ethical restrictions on sharing data publicly. 

b) If there are no restrictions, please upload the minimal anonymized data set necessary to replicate your study findings as either Supporting Information files or to a stable, public repository and provide us with the relevant URLs, DOIs, or accession numbers. 

Response: All data are presented in this manuscript or in the pan-European manuscripts previously. 

4. Thank you for stating the following in the Financial Disclosure section:

‘This study was sponsored by Aimmune Therapeutics, a Nestle Health Science company. Brainsell provided analytical support for this study. Editorial assistance and submission support were provided by The Curry Rockefeller Group, LLC (Tarrytown, NY); both were funded by Aimmune Therapeutics, a Nestle Health Science company.’ We note that one or more of the authors have an affiliation to the commercial funders of this research study: Aimmune Therapeutics. We also note that one or more of the authors are employed by other commercial companies: Acaster Lloyd Consulting, Brainsell Ltd. and Allergy UK. 

a. Please provide an amended Funding Statement declaring these commercial affiliations, as well as a statement regarding the Role of Funders in your study. If the funding organization did not play a role in the study design, data collection and analysis, decision to publish, or preparation of the manuscript and only provided financial support in the form of authors' salaries and/or research materials, please review your statements relating to the author contributions, and ensure you have specifically and accurately indicated the role(s) that these authors had in your study. You can update author roles in the Author Contributions section of the online submission form.

Response: The Author Contributions section was complete as included. We have added additional clarification regarding employment and the role of the funder as follows: “This work was supported by Aimmune Therapeutics. Authors Robert Ryan and Andrea Vereda are employed by Aimmune Therapeutics. Brainsell Ltd received funding from Aimmune Therapaeutics for conducting the APPEAL-1 survey and for analytical support. Author Ram Patel is employed by Brainsell. Acaster Lloyd Consulting received funding from Aimmune Therapeutics for conducting the study. Authors Katy Gallop and Sarah Acaster are employed by Acaster Lloyd Consulting. Allergy UK is a registered patient charity. The specific roles of the authors are articulated in the 'author contributions' section. The funders provided support in the form of salaries to these authors or consultant fees to the employers of these authors and were involved in the study design, data collection and analysis, the decision to publish, and preparation of the manuscript.” 

Response: We have updated the Funding statement as detailed in the request above follows: “This work was supported by Aimmune Therapeutics. Authors Robert Ryan and Andrea Vereda are employed by Aimmune Therapeutics. Brainsell Ltd received funding from Aimmune Therapaeutics for conducting the APPEAL-1 survey and for analytical support. Author Ram Patel is employed by Brainsell. Acaster Lloyd Consulting received funding from Aimmune Therapeutics for conducting the study. Authors Katy Gallop and Sarah Acaster are employed by Acaster Lloyd Consulting. The specific roles of these authors are articulated in the 'author contributions' section. The funders provided support in the form of salaries to these authors or consultant fees to the employers of these authors and were involved in the study design, data collection and analysis, the decision to publish, and preparation of the manuscript.” 

b. Please also provide an updated Competing Interests Statement declaring these commercial affiliations along with any other relevant declarations relating to employment, consultancy, patents, products in development, or marketed products, etc.

Within your Competing Interests Statement, please confirm that these commercial affiliations do not alter your adherence to all PLOS ONE policies on sharing data and materials by including the following statement: “This does not alter our adherence to PLOS ONE policies on sharing data and materials.” 

Response: We have updated the Competing Interests Statement and included the suggested statement as follows: “…The authors have read the journal’s policy and have the following competing interests: RR and AV are employed by Aimmune Therapeutics. SA, KG, and RP report consulting for Aimmune Therapeutics. MT, LR, AW, AD, and AB held consultancy and received sponsorship to attend and speak at educational events from pharmaceutical companies. This does not alter our adherence to PLOS ONE policies on sharing data and materials.”

Response: Please see responses above.

We appreciate the opportunity to revise our manuscript to address the peer-review comments. We hope you find our revised manuscript suitable for publication in PLOS ONE.

Sincerely,

Audrey DunnGalvin, MD

School of Applied Psychology and Department of Paediatrics and Child Health

University College Cork, Cork, Ireland 

Email: A.DunnGalvin@ucc.com

References

1. Blumchen K, DunnGalvin A, Timmermans F, Regent L, Schnadt S, Podesta M, et al. APPEAL-1: a pan-European survey of patient/caregiver perceptions of peanut allergy management. Allergy. 2020; 75(11): 2920-35.

---

## [Editor Report · Decision Letter 1]

7 Jan 2022

Allergy to Peanuts imPacting Emotions And Life (APPEAL): the impact of peanut allergy on children, teenagers, adults and caregivers in the UK and Ireland

PONE-D-20-40094R1

Dear Dr. Ryan,

We’re pleased to inform you that your manuscript has been judged scientifically suitable for publication and will be formally accepted for publication once it meets all outstanding technical requirements.

Kind regards,

Pei-Ning Wang

Academic Editor

PLOS ONE
---

## [Editor Report · Acceptance letter]

30 Jan 2022

PONE-D-20-40094R1 

Allergy to Peanuts imPacting Emotions And Life (APPEAL): the impact of peanut allergy on children, teenagers, adults and caregivers in the UK and Ireland 

Dear Dr. Ryan:

I'm pleased to inform you that your manuscript has been deemed suitable for publication in PLOS ONE. Congratulations! Your manuscript is now with our production department. 

Kind regards, 

on behalf of

Dr. Pei-Ning Wang 

Academic Editor

PLOS ONE